# Determinants of Kangaroo Mother Care among low-birth-weight infants in low resource settings

**Temitayo Victor Lawal** [1,2]*, **Damilola Israel Lawal** [1], **Oluwafemi John Adeleye** [1]

**1** Department of Epidemiology and Medical Statistics, Faculty of Public Health, University of Ibadan, Ibadan, Nigeria, **2** International Research Center of Excellence, Institute of Human Virology Nigeria, Abuja, FCT, Nigeria

* latevi24@gmail.com

## Abstract

Kangaroo Mother Care involves direct contact between a baby's bare skin and a caregiver, typically the mother. It has many benefits for both baby and caregiver and is often used to regulate body temperature, promote breastfeeding, enhance growth, and bonding. This study aims to explore factors associated with Kangaroo Mother Care uptake in low-resource countries for babies born with low-birth-weight. Demographic and Health Survey data from 34 low- and middle- income countries were analyzed. Cross-sectional data of 57,223 children were pooled and analyzed. Hierarchical multivariable analysis was performed to determine the factors associated with skin-to-skin contact. Statistical significance was set to 5%. The prevalence of Kangaroo Mother Care ranged from 11.04% to 84.36%; highest in Benin (84.36%), Tajikistan (80.88%), and Uganda (80.86%) and lowest in Burundi (11.04%), Bangladesh (16.58%), and Pakistan (19.24%). Higher odds of Kangaroo Mother Care were estimated among low-birth-weight infants who were put to breast immediately, had low-birth-weight ($\geq$1.5kg), born through normal delivery, born at health facility, those whose mothers were exposed to media, had high antenatal care visits, had formal education, and in the younger age bracket. Also, women living in communities with high illiteracy, countries in the lower-middle income region had higher odds of Kangaroo Mother Care. Women domiciled in Europe and Central Asia, Sub-Saharan Africa, Latin America and The Caribbeans, and East Asia and Pacific had lower odds of Kangaroo Mother Care. This study found a low uptake of Kangaroo Mother Care in countries with limited resources, which is a concerning issue that requires urgent attention. Increasing awareness, education, and support for mothers and families to practice Kangaroo Mother Care, as well as training healthcare practitioners, can lead to better outcomes for newborns and reduce neonatal death.

## Introduction

Across the globe, about 8.2million children under the age of 5 die each year, out of which 2.4million of these deaths are reported in the first four weeks of life [1]. Reducing neonatal

**Data Availability Statement:** The data used for this study are made publicly available to researchers, policymakers, and other interested parties through the DHS Program website (www.dhsprogram.com).

**Funding:** The authors received no specific funding for this work.

**Competing interests:** The authors have declared that no competing interests exist.

death, which is usually caused by three major conditions: asphyxia, infection and prematurity, is a global aim under the Sustainable Development Goal [2]. The prevalence of neonatal deaths is higher among children born preterm and/or with low-birth-weight (LBW) and these LBW infants who survive the neonatal period are more likely to experience neonatal morbidities including acute respiratory, gastrointestinal, immunologic, and central nervous system problems than children born term and with normal weights [3].

Previous researchers have reported that infants having LBW usually have more adverse effect on the survival and development of neonates [4–6]. It is therefore recommended that all babies born should receive essential newborn care to reduce the risk of neonatal morbidities and/or death, including thermal protection (often referred to as Kangaroo Mother Care or skin-to-skin contact), hygienic umbilical cord and skin care, preventive treatment, and early and exclusive breastfeeding [1, 7, 8].

Kangaroo Mother Care (KMC) or skin-to-skin contact between mother and baby is a multi-faceted intervention for preterm and LBW infants that plays a significant role on infant survival, and quality of mother-infant bonding by providing inexpensive but quality care in place of warmers and incubators [9, 10]. A 20-year follow-up study carried out to evaluate the importance of KMC against traditional care among LBW infants found that KMC had a significant and long-lasting social and behavioral protective effect on the babies [10].

Furthermore, KMC initiates early-maternal attachment and helps preterm infants stabilize physiological symptoms, while reducing maternal stress level, risk of sepsis and postpartum hospital stay [4, 8]. Further studies have also found that the practice of KMC helps infants receive higher mother's milk during hospitalization, increases the chance of exclusive breast-feeding, and the chances of improved body features, such as body weight, body length and head circumference, at follow-up [11, 12].

Despite the availability of studies in the literature on KMC for newborn infants across low- and middle-income countries (LMICs), there is still a gap in knowledge about existing variation in practice of KMC across multiple countries and regions of the world. This remains a grey area for research and this study intend to fill the gap. Also, studies that examined the drivers of practice of KMC among infants born with LBW is scarce in the literature. In light of this, this study employed a robust logistics multilevel modelling strategy that measures the impact of individual-, community- and country-level differences in the practice of KMC across LMICs, to determine the determinants of KMC practice. The aim of this study is to examine the practice of KMC among LBW infants in LMICs. This would be useful for policy makers and other researchers to identify areas where KMC needs to be strengthened to improve newborn care.

## Materials and methods

### Study design and data

The data used in this study was a cross-sectional and nationally representative survey of the Demographic and Health Survey (DHS). This data was collected across most LMICs. For the purpose of analysis, we extracted and pooled the latest recode of the children's data from the DHS, as at March 2023. The DHS used a multi-stage, stratified sampling design with clusters as the primary sampling units, and households as sampling units. Clusters (enumeration areas) were selected from already identified rural and urban local government areas, but the number of sampling stages differed by country–due to the irregularities in the administrative levels across the countries. Sampling weights were added to account for unequal probability of selection at the cluster levels and non-response since the samples were not self-weighting. These weights helped to minimize non-response and selection biases. All the questionnaires

used across the countries were standardized and deployed with similar implantation protocols. For further information on the country-specific methods employed in DHS, see www. dhsprogram.com.

### Data source

The secondary data used for this study is available on request from the owners at https://www. dhsprogram.com/data/dataset_admin/login_main.cfm. The DHS survey was conducted in 91 countries, out of which 84 were domiciled in LMICs. For the purpose of this study, 34 countries with data on KMC were extracted and pooled for analysis.

### Dependent variable

The dependent variable considered in this study is the practice of KMC at birth among LBW infants (children born with birth weight < 2.5kg). Information on whether the child was put on mother's bare skin after birth was asked. The response was recoded as "1" if the child was put on the mother's bare skin after birth, and "0" otherwise.

### Explanatory variables

This study used three levels of explanatory variables based on findings in the literature, and availability of data. Hierarchical levels of explanatory variables captured individual, community, and country levels.

### Individual-level factors

Factors at this level are characteristics of the child (including sex and birth weight) and mother (including mother's age, educational level, marital status, place of delivery, number of antenatal care visits, total children ever born, mother's use of internet, exposure to media, wealth index (derived by DHS as a proxy for socioeconomic status), mode of delivery, and when child was put to breast). Exposure to media, in this study, was defined as the mother's access to information through any of newspapers/magazines, radio or television (i.e., if the mother reads or watches any, at least once a week). Birth weight was also recategorized as extremely LBW (<1kg), very LBW (<1.5kg), and LBW (<2.5kg).

### Community-level factors

Variables considered in the community level include place of residence, community illiteracy and community poverty. Community in this study was defined as clustering of individuals within the same geographical living area, and sharing the same primary sampling unit within DHS.

### Country-level factors

Country-level factors were retrieved from data published by the world bank on development indicator. The countries were grouped according to the World Bank income classification and regional location at the year of the survey implementation. Specific details can be found at https://data.worldbank.org/income-level/low-and-middle-income and https://datatopics. worldbank.org/world-development-indicators/stories/the-classification-of-countries-by-income.html.

## Data management and analysis

Available data of LMICs was extracted and pooled from the DHS website. Data cleaning was carried out to handle missingness and recoding. Individuals without record on practice of KMC were dropped from the data. Prior to data analysis, information of children with LBW were retained, and a weighted univariate and bivariate analysis was done. Sampling weights provided by DHS were applied to country-level data to adjust for unequal cluster sizes, stratifications, and ensure national representation.

Prior to the inferential analysis, multicollinearity test was performed among the independent variables, and variables with variance inflation factor > 10 were dropped from the analysis. All analysis were carried out in STATA version 16.

## Statistical modelling

For the purpose of statistical modelling, a multivariable multilevel logistic regression model was employed to identify factors that are associated with the practice of KMC among LBW weight infants in LMICs. The model employed was hierarchical in nature with mixed outcomes comprising of fixed and random parts:

$$logit(\pi_{ijk}) = \beta_o + \sum_{p=1}^{p} \beta_p X_{pijk} + U_{0jk} + V_{0k}$$

Where $\beta_o$ is the intercept; $\beta_p$ is the regression coefficient for the p parameters, $X_{pijk}$ are the covariates, $U_{0jk}$ is the random component attributable to children in community j of country k, and $V_{0k}$ is the random component attributable to all children in country k.

We developed five distinct models to consider different combination of factors that may be suitable for the data. The most robust of the models that could identify the risk factors for the practice of KMC among LBW infants in LMICS was selected using the Bayesian Information Criterion (BIC) and/or Akaike Information Criterion (AIC). The first model was the null model that assessed the variation due to community and country random effects without incorporating any explanatory variable. The second model included only individual-level variables conditional on community and country-level effects. Similar to the second model, the third and fourth model examined the community and country-level characteristics, while the final model estimated all the variables across levels.

## Ethical approval

The study was based on the analysis of openly available secondary data. The authors did not have access to information that could identify individual participants during or after data collection. Ethical approvals were obtained from the Ethics Committee of the ICF Macro at Fairfax, Virginia in the USA and by the National Ethics Committees in Nigeria. Prior to commencement of data collection, written and signed informed consent was obtained from each parent and/or legal guardians of the children who participated in the study. The study posed minimal risk to the participants, and all information was collected anonymously and held confidentially. All methods were carried out in accordance with relevant guidelines and regulations.

## Results

A total of 57,223 infants born with LBW were pooled from the survey of 34 LMICs across the world (Fig 1). Findings from the study revealed that 68.8% of the infants experienced KMC, touching bare skin of mother (Table 1). Practice of KMC was highest among children whose mothers were aged 20–24 and 25–29 years– 70.7% and 70.9% respectively. The practice of

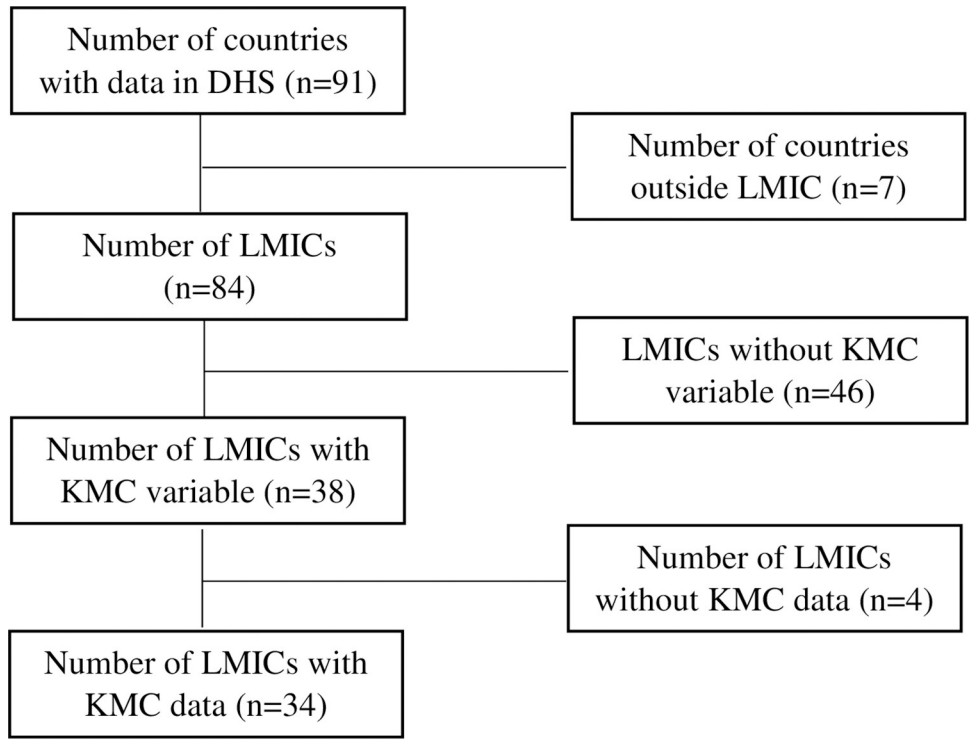

**Fig 1. CONSORT diagram.**

KMC increased with level of education but experienced a reduction for children whose mothers had tertiary education. Notably, the practice of KMC was highest among LBW infants who were put to breast immediately after birth. On the other hand, practice of KMC was lowest among infants with extremely LBW (<1000kg).

Fig 2 presents the country-specific practice of KMC among LBW infants. The practice of KMC across countries ranged from 11.04% to 84.36%. It is worthy to note that Burundi had the lowest practice of KMC, followed by Bangladesh and Pakistan. On the flip side the practice of KMC among LBW infants was highest in Benin, followed by Uganda and Tajikistan. In Table 2, the estimates and information criterion of the five-models considered for this study was presented. Based on estimates from the AIC and BIC, the fully adjusted model (Model V) used to control for the effects of all the aforementioned levels (individual-, community, and country-level factors) had the lowest information criterion, and therefore was selected as the most suitable for the model.

The study observed a significant variation in the practice of KMC among LBW infants across countries. The confidence interval of the variance for country level estimate was statistically significant (variance significantly different from zero), so the hypothesis that the regression slopes for the practice of KMC vary across the countries is supported by the data. The intra-country correlation coefficient (ICC) indicates that about 50% of the variation in KMC practice was due to country level differences, whereas the remaining 50% was attributable to individual- and community-level differences.

The odds of KMC practice increased with mother's age–the odds of KMC practice were significantly higher among infants whose mothers are aged 20–39 years than women aged 15–19 years (Table 3). Furthermore, education was also a statistically significant predictor of KMC practice–the odds of KMC practice was higher among infants whose mother had primary/

**Table 1. Weighted distribution of respondent's characteristics by practice of KMC at birth.**

| | Total | KMC done touching bare skin [n (row %)] | KMC done but no touching of bare skin [n (row %)] | KMC done but touching of bare skin missing [n (row %)] | KMC not done [n (row %)] | $\chi^2$ p-value |
|---|---|---|---|---|---|---|
| **Overall** | **57,223** | **39,341 (68.8)** | **1,871 (3.3)** | **84 (0.2)** | **15,927 (27.8)** | |
| **Level 1 (Individual Level Characteristics)** | | | | | | |
| **Age group** | | | | | | |
| 15–19 years | 2,616 | 1,707 (65.2) | 74 (2.9) | 4 (0.1) | 832 (31.8) | < 0.001 * |
| 20–24 years | 17,229 | 12,180 (70.7) | 612 (3.6) | 20 (0.1) | 4,417 (25.6) | |
| 25–29 years | 19,893 | 14,096 (70.9) | 674 (3.4) | 35 (0.2) | 5,088 (25.6) | |
| 30–34 years | 10,281 | 6,784 (66.9) | 313 (3.0) | 22 (0.2) | 3,072 (29.9) | |
| 35–39 years | 5,067 | 3,246 (64.1) | 148 (2.9) | 1 (0.0) | 1,672 (33.0) | |
| 40–44 years | 1,755 | 1,016 (57.9) | 43 (2.5) | 2 (0.1) | 694 (39.5) | |
| 45–49 years | 381 | 222 (58.3) | 7 (1.9) | - | 152 (39.8) | |
| **Mother's highest level of education** | | | | | | |
| No Formal Education | 11,460 | 7,865 (68.6) | 388 (3.4) | 16 (0.1) | 3,192 (27.9) | < 0.001 * |
| Primary/Basic Education | 11,512 | 7,690 (66.8) | 326 (2.8) | 11 (0.1) | 3,484 (30.3) | |
| Secondary Education | 27,259 | 19,196 (70.4) | 973 (3.6) | 48 (0.2) | 7,042 (25.8) | |
| Tertiary Education | 6,992 | 4,590 (65.7) | 184 (2.6) | 9 (0.1) | 2,209 (31.6) | |
| **Mother's marital status** | | | | | | |
| Living alone | 3,026 | 1,814 (60.0) | 60 (2.0) | 6 (0.2) | 1,145 (37.8) | < 0.001 * |
| Living with partner | 54,197 | 37,527 (69.2) | 1,811 (3.3) | 78 (0.1) | 14,782 (27.3) | |
| **Place of delivery** | | | | | | |
| Home or elsewhere | 4,502 | 2,666 (59.2) | 183 (4.1) | 9 (0.2) | 1,644 (36.5) | < 0.001 * |
| Public facility | 39,067 | 28,202 (72.2) | 1,265 (3.2) | 56 (0.1) | 9,544 (24.4) | |
| Private facility | 13,654 | 8,473 (62.1) | 423 (3.1) | 19 (0.1) | 4,740 (34.7) | |
| **Number of ANC visits** | | | | | | |
| No ANC visit | 17,506 | 12,047 (68.8) | 568 (3.2) | 27 (0.2) | 4,864 (27.8) | 0.003 * |
| 1–3 visits | 20,817 | 14,340 (68.9) | 729 (3.5) | 30 (0.1) | 5,718 (27.5) | |
| 4–7 visits | 10,838 | 7,312 (67.5) | 348 (3.2) | 12 (0.1) | 3,166 (29.2) | |
| 8 visits and above | 8,062 | 5,641 (70.0) | 226 (2.8) | 15 (0.2) | 2,180 (27.0) | |
| **Total children ever born** | | | | | | |
| One | 14,843 | 10,122 (68.2) | 475 (3.2) | 20 (0.1) | 4,226 (28.5) | < 0.001 * |
| Two | 19,791 | 14,016 (70.8) | 620 (3.1) | 33 (0.2) | 5,122 (25.9) | |
| Three | 10,645 | 7,412 (69.6) | 405 (3.8) | 17 (0.2) | 2,810 (26.4) | |
| Four and above | 11,944 | 7,791 (65.2) | 371 (3.1) | 14 (0.1) | 14 (31.6) | |
| **Mother's use of internet** | | | | | | |
| Never used internet | 17,794 | 10,824 (60.8) | 428 (2.4) | 6 (0.0) | 6,536 (36.7) | < 0.001 * |
| Ever used internet | 6,980 | 4,368 (62.6) | 277 (4) | 20 (0.3) | 2,314 (33.2) | |
| Missing | 32,449 | 24,149 (74.4) | 1,166 (3.6) | 58 (0.2) | 7,077 (21.8) | |
| **Exposure to media** | | | | | | |
| Not exposed | 41,002 | 28,392 (69.3) | 1,422 (3.5) | 59 (0.1) | 11,129 (27.1) | < 0.001 * |
| Exposed | 16,221 | 10,949 (67.5) | 449 (2.8) | 25 (0.2) | 4,798 (29.6) | |
| **Wealth Index** | | | | | | |
| Poor | 26,054 | 18,658 (71.6) | 980 (3.8) | 47 (0.2) | 6,368 (24.4) | < 0.001 * |
| Average | 11,207 | 7,691 (68.6) | 319 (2.9) | 9 (0.1) | 3,188 (28.4) | |
| Rich | 19,962 | 12,992 (65.1) | 571 (2.9) | 28 (0.1) | 6,371 (31.9) | |
| **Sex of child** | | | | | | |

*(Continued)*

**Table 1.**  (Continued)

| | Total | KMC done touching bare skin [n (row %)] | KMC done but no touching of bare skin [n (row %)] | KMC done but touching of bare skin missing [n (row %)] | KMC not done [n (row %)] | $\chi^2$ p-value |
|---|---|---|---|---|---|---|
| **Overall** | **57,223** | **39,341 (68.8)** | **1,871 (3.3)** | **84 (0.2)** | **15,927 (27.8)** | |
| **Level 1 (Individual Level Characteristics)** | | | | | | |
| Male | 27,384 | 18,708 (68.3) | 896 (3.3) | 54 (0.2) | 7,726 (28.2) | 0.022 * |
| Female | 29,839 | 20,633 (69.2) | 975 (3.3) | 30 (0.1) | 8,201 (27.5) | |
| **Mode of delivery** | | | | | | |
| Normal birth | 45,708 | 33,183 (72.6) | 1,592 (3.5) | 59 (0.1) | 10,873 (23.8) | < 0.001 * |
| Caesarean section | 11,425 | 6,093 (53.3) | 275 (2.4) | 25 (0.2) | 5,032 (44.1) | |
| Missing | 90 | 65 (71.9) | 3 (3.6) | - | 22 (24.5) | |
| **Birth weight of child** | | | | | | |
| Extremely low birth weight | 767 | 389 (50.8) | 28 (3.7) | 2 (0.2) | 347 (45.3) | < 0.001 * |
| Very low birth weight | 3,744 | 2,175 (58.1) | 138 (3.7) | 6 (0.2) | 1,425 (38.1) | |
| Low birth weight | 52,712 | 36,777 (69.8) | 1,704 (3.2) | 76 (0.2) | 14,155 (26.9) | |
| **When child was put to breast** | | | | | | |
| Immediately | 25,006 | 19,016 (76.1) | 751 (3.0) | 27 (0.1) | 5,212 (20.8) | < 0.001 * |
| Within 24 hours | 22,651 | 15,801 (69.8) | 809 (3.6) | 35 (0.2) | 6,007 (26.5) | |
| 24 hours or more | 6,050 | 2,728 (45.1) | 173 (2.9) | 10 (0.2) | 3,139 (51.9) | |
| Missing | 3,516 | 1,796 (51.1) | 138 (3.9) | 12 (0.3) | 1,570 (44.7) | |
| **Level 2 (Community Level Characteristics)** | | | | | | |
| **Place of residence** | | | | | | |
| Urban | 18,183 | 11,950 (65.7) | 526 (2.9) | 21 (0.1) | 5,687 (31.3) | < 0.001 * |
| Rural | 39,040 | 27,391 (70.2) | 1,345 (3.4) | 63 (0.2) | 10,240 (26.2) | |
| **Community Illiteracy** | | | | | | |
| Low | 26,428 | 19,101 (72.3) | 887 (3.4) | 49 (0.2) | 6,390 (24.2) | < 0.001 * |
| Average | 12,980 | 7,883 (60.7) | 363 (2.8) | 7 (0.1) | 4,726 (36.4) | |
| High | 17,815 | 12,357 (69.4) | 620 (3.5) | 28 (0.2) | 4,810 (27.0) | |
| **Community Poverty** | | | | | | |
| Low | 24,140 | 17,411 (72.1) | 837 (3.5) | 43 (0.2) | 5,849 (24.2) | < 0.001 * |
| Average | 15,476 | 9,060 (58.6) | 414 (2.7) | 13 (0.1) | 5,989 (38.7) | |
| High | 17,607 | 12,869 (73.1) | 620 (3.5) | 28 (0.2) | 4,089 (23.2) | |
| **Level 3 (Country Level Characteristics)** | | | | | | |
| **Country Income** | | | | | | |
| Low income | 6,731 | 3,581 (53.2) | 78 (1.2) | 2 (0.0) | 3,070 (45.6) | < 0.001 * |
| Lower middle income | 47,345 | 34,013 (71.8) | 1,565 (3.3) | 77 (0.2) | 11,689 (24.7) | |
| Upper middle income | 3,147 | 1,747 (55.5) | 228 (7.3) | 5 (0.2) | 1,167 (37.1) | |
| **Region** | | | | | | |
| Middle East and North Africa | 1,674 | 991 (59.2) | 178 (10.6) | 4 (0.3) | 501 (29.9) | < 0.001 * |
| South Asia | 39,139 | 28,907 (73.9) | 1,386 (3.5) | 70 (0.2) | 8,777 (22.4) | |
| Europe and Central Asia | 1,307 | 726 (55.6) | 34 (2.6) | - | 547 (41.8) | |
| Sub-Saharan Africa | 11,425 | 6,439 (56.4) | 178 (1.6) | 4 (0.0) | 4,805 (42.1) | |
| Latin America and The Caribbeans | 342 | 187 (54.7) | 20 (5.9) | - | 135 (39.4) | |
| East Asia and Pacific | 3,336 | 2,092 (62.7) | 75 (2.2) | 6 (0.2) | 1,163 (34.9) | |

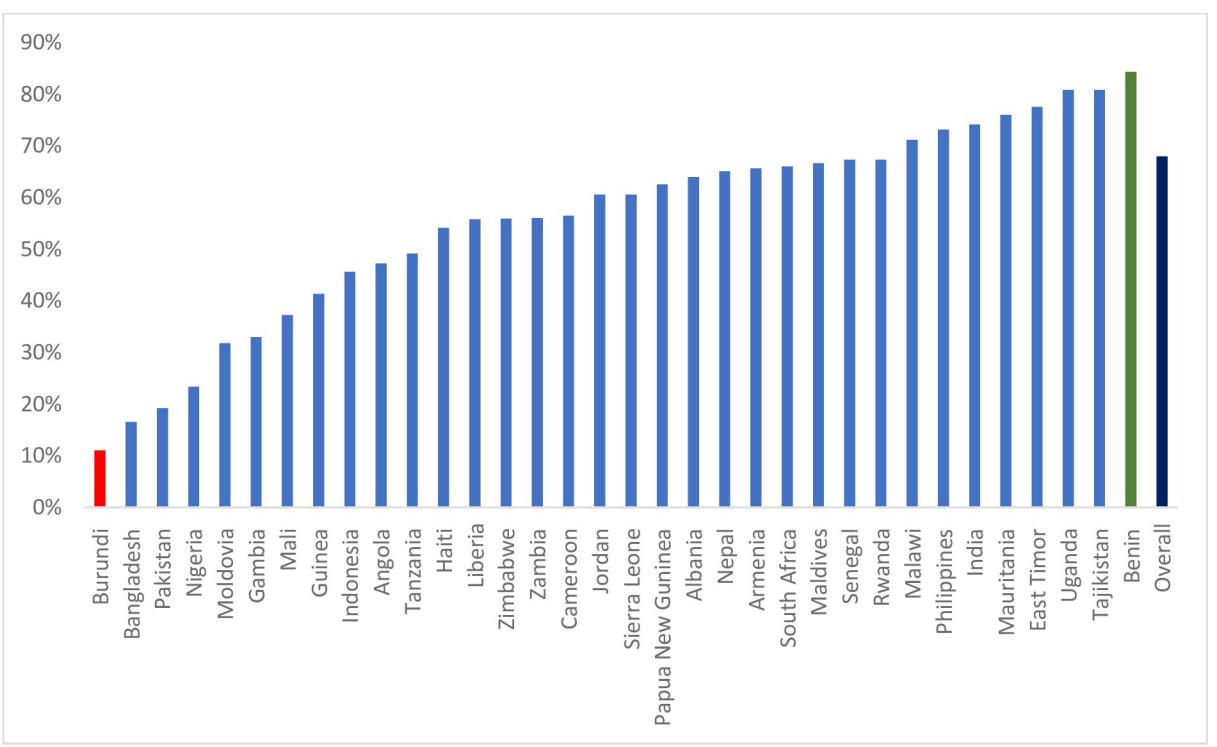

**Fig 2. Practice of KMC among LBW infants in LMICs.**

basic, secondary, and tertiary education than infants whose mothers had no formal education (Odds Ratio (OR) = 1.50, 1.42, 1.65; 95% Confidence Interval (CI): 1.32–1.71, 1.24–1.71, 1.34–2.03 respectively).

The odds of KMC practice were higher among infants who were delivered at public facilities (OR: 4.87, 95% CI: 4.19–5.67) and private facilities (OR: 3.45; 95% CI: 2.87–4.15) than infants who were delivered at home or elsewhere. Similarly, infants whose mothers attended 8 or more antenatal care visits had a significantly higher odds of KMC practice (OR: 1.27; 95% CI: 1.10–1.47). On the contrary, infants who were delivered through caesarean section had lower odds of KMC than infants who were delivered through normal delivery (OR: 0.22; 95% CI: 0.20–0.26). Similarly, infants who were put to breast within 24hours of life and infants who were put to breast later than 24hours of life had lower odds of KMC than infants who were put to breast immediately (OR: 0.50, 0.20; 95% CI: 0.50–0.61, 0.17–0.23 respectively).

The lower the birth weight, the lower the odds of KMC–infants who were born with LBW (<2.5kg) had higher odds of KMC than infants born with an extremely LBW (<1kg). Also, infants whose mothers were exposed to media had a higher odds of KMC practice at birth (OR: 1.28; 95% CI: 1.16–1.41).

From the community-level factors, infants whose mothers reside in communities with high illiteracy had higher odds of KMC than infants whose mothers reside in communities with low community illiteracy. Estimates from the country-level factors also revealed that the odds of KMC was higher among infants born in lower middle-income countries than low-income countries (OR: 1.81; 95% CI: 1.55–2.11). Infants also born in South Asia, Europe and Central Asia, sub-Saharan Africa, Latin America and the Caribbeans, and East Asia and Pacific had lower odds of KMC than children born in the Middle East and North Africa.

**Table 2. Logit three-level variance component model for the practice of KMC among mothers of low-birth-weight infants in LMICs.**

| | Model I β (S.E.) | Model II β (S.E.) | Model III β (S.E.) | Model IV β (S.E.) | Model V β (S.E.) |
|---|---|---|---|---|---|
| **FIXED EFFECTS** | | | | | |
| **Level 1 (Individual Level Characteristics)** | | | | | |
| **Age group** | | | | | |
| 15–19 years | | Reference | | | Reference |
| 20–24 years | | 0.32 (0.10) | | | 0.25 (0.10) |
| 25–29 years | | 0.42 (0.10) | | | 0.29 (0.10) |
| 30–34 years | | 0.34 (0.11) | | | 0.23 (0.11) |
| 35–39 years | | 0.33 (0.12) | | | 0.25 (0.12) |
| 40–44 years | | 0.04 (0.14) | | | - 0.03 (0.14) |
| 45–49 years | | 0.10 (0.21) | | | 0.04 (0.21) |
| **Mother's highest level of education** | | | | | |
| No Formal Education | | Reference | | | Reference |
| Primary/Basic Education | | 0.25 (0.06) | | | 0.41 (0.07) |
| Secondary Education | | 0.44 (0.07) | | | 0.35 (0.07) |
| Tertiary Education | | 0.62 (0.10) | | | 0.50 (0.11) |
| **Mother's marital status** | | | | | |
| Living alone | | Reference | | | Reference |
| Living with partner | | 0.33 (0.07) | | | 0.07 (0.07) |
| **Place of delivery** | | | | | |
| Home or elsewhere | | Reference | | | Reference |
| Public facility | | 1.49 (0.08) | | | 1.58 (0.08) |
| Private facility | | 1.30 (0.09) | | | 1.24 (0.09) |
| **Number of ANC visits** | | | | | |
| No ANC visit | | Reference | | | Reference |
| 1–3 visits | | 0.07 (0.05) | | | 0.09 (0.05) |
| 4–7 visits | | 0.08 (0.06) | | | 0.08 (0.06) |
| 8 visits and above | | 0.33 (0.07) | | | 0.24 (0.08) |
| **Total children ever born** | | | | | |
| One | | Reference | | | Reference |
| Two | | 0.02 (0.07) | | | 0.02 (0.06) |
| Three | | - 0.12 (0.08) | | | - 0.06 (0.08) |
| Four and above | | - 0.14 (0.08) | | | - 0.00 (0.08) |
| **Mother's use of internet** | | | | | |
| Never used internet | | Reference | | | Reference |
| Ever used internet | | 0.04 (0.06) | | | - 0.01 (0.06) |
| **Exposure to media** | | | | | |
| Not exposed | | Reference | | | Reference |
| Exposed | | 0.15 (0.05) | | | 0.25 (0.05) |
| **Wealth Index** | | | | | |
| Poor | | Reference | | | Reference |
| Average | | - 0.10 (0.06) | | | - 0.03 (0.06) |
| Rich | | - 0.24 (0.06) | | | - 0.20 (0.06) |
| **Sex of child** | | | | | |
| Male | | Reference | | | Reference |
| Female | | 0.03 (0.04) | | | 0.04 (0.04) |
| **Mode of delivery** | | | | | |

(*Continued*)

**Table 2.** (Continued)

| | Model I β (S.E.) | Model II β (S.E.) | Model III β (S.E.) | Model IV β (S.E.) | Model V β (S.E.) |
|---|---|---|---|---|---|
| Normal birth | | Reference | | | Reference |
| Caesarean section | | - 1.38 (0.07) | | | - 1.49 (0.07) |
| **Birth weight of child** | | | | | |
| Extremely low birth weight | | Reference | | | Reference |
| Very low birth weight | | 0.15 (0.17) | | | 0.10 (0.17) |
| Low birth weight | | 0.49 (0.16) | | | 0.44 (0.16) |
| **When child was put to breast** | | | | | |
| Immediately | | Reference | | | Reference |
| Within 24 hours | | - 0.47 (0.05) | | | - 0.59 (0.05) |
| 24 hours or more | | - 1.54 (0.08) | | | - 1.60 (0.08) |
| **Level 2 (Community Level Characteristics)** | | | | | |
| **Place of residence** | | | | | |
| Urban | | | Reference | | Reference |
| Rural | | | 0.24 (0.04) | | - 0.03 (0.06) |
| **Community Illiteracy** | | | | | |
| Low | | | Reference | | Reference |
| Average | | | - 0.25 (0.06) | | 0.13 (0.09) |
| High | | | 0.10 (0.04) | | 0.18 (0.09) |
| **Community Poverty** | | | | | |
| Low | | | Reference | | Reference |
| Average | | | - 0.75 (0.06) | | - 0.19 (0.10) |
| High | | | - 0.04 (0.04) | | - 0.03 (0.09) |
| **Level 3 (Country Level Characteristics)** | | | | | |
| **Country Income** | | | | | |
| Low income | | | | Reference | Reference |
| Lower middle income | | | | 0.36 (0.07) | 0.59 (0.08) |
| Upper middle income | | | | - 0.20 (0.12) | 0.11 (0.14) |
| **Region** | | | | | |
| Middle East and North Africa | | | | Reference | Reference |
| South Asia | | | | 0.39 (0.13) | 0.16 (0.15) |
| Europe and Central Asia | | | | - 0.36 (0.14) | - 0.91 (0.16) |
| Sub-Saharan Africa | | | | - 0.43 (0.13) | - 0.77 (0.15) |
| Latin America and The Caribbeans | | | | - 0.87 (0.23) | - 1.15 (0.26) |
| East Asia and Pacific | | | | - 0.38 (0.14) | - 0.57 (0.16) |
| **Model Fit Statistics (Information Criterion)** | | | | | |
| - 2 Log Likelihood | - 33115.02 | - 13805.34 | - 32852.20 | - 32759.73 | - 13569.08 |
| AIC | 66236.04 | 27.672.68 | 65720.40 | 65537.46 | 27222.15 |
| BIC | 66262.85 | 27922.61 | 65791.91 | 65617.9 | 27560.76 |

## Discussion

In resource-limited countries, preterm babies have a high mortality rate, and they are at risk of a range of health problems such as, respiratory distress syndrome, infections, intraventricular hemorrhage, necrotizing enterocolitis, and hypothermia, which leads to serious complications and in adverse situations, death [13–15]. In the face of the multifaceted challenges surrounding child-birth and preterm infant health, KMC practice has been proven to be effective in strengthening immunity, increase psychological function and longer-term neurodevelopment

**Table 3. Logit three-level regression model for the practice of KMC among mothers of low-birth-weight infants in LMICs.**

|  | Odds Ratio | 95% Confidence Interval |
|---|---|---|
| **FIXED EFFECTS** | | |
| **Level 1 (Individual Level Characteristics)** | | |
| **Age group** | | |
| 15–19 years | Reference | |
| 20–24 years | 1.28 | 1.06–1.55 |
| 25–29 years | 1.33 | 1.09–1.63 |
| 30–34 years | 1.25 | 1.01–1.56 |
| 35–39 years | 1.29 | 1.02–1.63 |
| 40–44 years | 0.97 | 0.74–1.27 |
| 45–49 years | 1.04 | 0.79–1.57 |
| **Mother's highest level of education** | | |
| No Formal Education | Reference | |
| Primary/Basic Education | 1.50 | 1.32–1.71 |
| Secondary Education | 1.42 | 1.24–1.64 |
| Tertiary Education | 1.65 | 1.34–2.03 |
| **Mother's marital status** | | |
| Living alone | Reference | |
| Living with partner | 1.08 | 0.94–1.24 |
| **Place of delivery** | | |
| Home or elsewhere | Reference | |
| Public facility | 4.87 | 4.19–5.67 |
| Private facility | 3.45 | 2.87–4.15 |
| **Number of ANC visits** | | |
| No ANC visit | Reference | |
| 1–3 visits | 1.10 | 0.99–1.21 |
| 4–7 visits | 1.08 | 0.96–1.21 |
| 8 visits and above | 1.27 | 1.10–1.47 |
| **Total children ever born** | | |
| One | Reference | |
| Two | 1.02 | 0.90–1.16 |
| Three | 0.94 | 0.81–1.09 |
| Four and above | 1.00 | 0.85–1.17 |
| **Mother's use of internet** | | |
| Never used internet | Reference | |
| Ever used internet | 0.99 | 0.87–1.11 |
| **Exposure to media** | | |
| Not exposed | Reference | |
| Exposed | 1.28 | 1.16–1.41 |
| **Wealth Index** | | |
| Poor | Reference | |
| Average | 0.97 | 0.86–1.09 |
| Rich | 0.91 | 0.81–1.03 |
| **Sex of child** | | |
| Male | Reference | |
| Female | 1.05 | 0.97–1.13 |
| **Mode of delivery** | | |

*(Continued)*

**Table 3.** (Continued)

|  | Odds Ratio | 95% Confidence Interval |
|---|---|---|
| Normal birth | Reference | |
| Caesarean section | 0.22 | 0.20–0.26 |
| **Birth weight of child** | | |
| Extremely low birth weight | Reference | |
| Very low birth weight | 1.11 | 0.79–1.56 |
| Low birth weight | 1.55 | 1.14–2.10 |
| **When child was put to breast** | | |
| Immediately | Reference | |
| Within 24 hours | 0.55 | 0.50–0.61 |
| 24 hours or more | 0.20 | 0.17–0.23 |
| **Level 2 (Community Level Characteristics)** | | |
| **Place of residence** | | |
| Urban | Reference | |
| Rural | 0.97 | 0.86–1.09 |
| **Community Illiteracy** | | |
| Low | Reference | |
| Average | 1.14 | 0.96–1.34 |
| High | 1.20 | 1.01–1.42 |
| **Community Poverty** | | |
| Low | Reference | |
| Average | 0.83 | 0.68–1.00 |
| High | 0.97 | 0.81–1.17 |
| **Level 3 (Country Level Characteristics)** | | |
| **Country Income** | | |
| Low income | Reference | |
| Lower middle income | 1.81 | 1.55–2.11 |
| Upper middle income | 1.11 | 0.86–1.45 |
| **Region** | | |
| Middle East and North Africa | Reference | |
| South Asia | 1.17 | 0.87–1.58 |
| Europe and Central Asia | 0.40 | 0.29–0.55 |
| Sub-Saharan Africa | 0.46 | 0.34–0.62 |
| Latin America and The Caribbeans | 0.32 | 0.19–0.52 |
| East Asia and Pacific | 0.57 | 0.41–0.78 |
| **RANDOM EFFECTS** | | |
| $\sigma_u^2$ | - | - |
| $\sigma_v^2$ | 3.32 | 2.97–3.70 |
| ICC ($U_\rho$) | - | - |
| ICC ($V_\rho$) | 0.50 | 0.47–0.53 |

for premature/LBW babies [4, 16, 17]. Given the limited research on KMC in low resource settings, this study thus examined the factors influencing KMC uptake among LBW infants.

The uptake of KMC ranged between 11.0% and 83.5%. Findings from this study reiterates that the KMC uptake for LBW infants is low especially in low-resource settings (considering the fact that the practice has been in existence for almost 4 decades) [18, 19]. Previous researchers have further reported that the practice keeps declining, despite the World Health Organization's (WHO) recommendation that healthy mothers and newborns, regardless of delivery or feeding

method, should have uninterrupted skin-to-skin contact [20, 21]. This decline has been attributed to low awareness of KMC, and its benefits for both the mother and child [22, 23].

This study reveals that the KMC practice was highest in Benin, followed by Uganda, and Tajikistan. Conversely, practice was lowest in Burundi, Bangladesh, and Pakistan. This finding supports that the variation in KMC may be attributed to differences between countries. This finding aligns with previous studies that attributed difference in health service uptake to cultural and societal differences, low level of awareness and country healthcare policies [3, 4, 22]. The pronounced uptake of KMC in some parts of East and West Africa may imply that mothers have substantial knowledge about the benefits of early breastfeeding and skin-to-skin care for their newborn [19, 24].

## Individual-level factors

An increase in mother's age was found to significantly increase the uptake of KMC among LBW infants. Previous researchers have reported a correlation between age, maternal experience, and the utilization of health services [25, 26]. Older mothers may have had more maternal and childbirth experience potentially through previous pregnancies or having cared for infants within their families or communities, which may in turn impact their uptake of health services. Other studies have reported that KMC was higher among older women compared to their younger counterparts [4, 9, 27].

Similarly, women with higher education had higher odds of KMC practice. Education plays a crucial role in imparting knowledge and raising awareness about important health practices, including KMC [28–30]. Educated mothers often have more health information, better access to healthcare, and an understanding of medical concepts. This knowledge and awareness, which may also be acquired through media exposure, can positively influence their recognition of the benefits and importance of KMC, potentially leading to a higher likelihood of adopting and practicing it [4, 31]. Furthermore, this exposure may cause educated mothers to challenge traditional practices that are not in line with evidence-based recommendations, such as separating premature infants from their mothers.

Delivery in a health facility (including private and public centres) increases the likelihood of KMC initiation and practice. Health facilities often have trained healthcare professionals who are knowledgeable about KMC and can provide guidance and support to mothers in practicing it effectively compared to home births which are often characterized by unskilled personnel and a lack of medical equipment [22, 32]. KMC can be initiated soon after birth, and healthcare staff can provide education and demonstrate the proper techniques to mothers. Similar to a finding in Ethiopia, practice of KMC was also found to be higher in Public than Private health facilities–this can be attributed to strict observance of national KMC guidelines in public facilities than private facilities who are not subject to strict public health sector regulation [19, 25]. Notably, ANC visit significantly improved KMC uptake. Mothers who attend up to 8 antenatal clinics (as recommended by WHO) may have a higher tendency of health facility delivery compared to those who never attended or attended a few [33, 34].

Furthermore, this study shows that a child's mode of delivery could impact KMC. Children born through normal delivery had lower odds of KMC compared to children born through normal delivery. This is likely due to delays caused by extended recovery time, as well as the postoperative pain experienced by mothers that restricts their mobility and limits access to the baby's bed [12, 22]. By extension, KMC may help early initiation of breastfeeding. KMC and breastfeeding are closely intertwined and mutually beneficial practices for the health and well-being of newborns, particularly LBW infants, by prompting the initiation of breastfeeding due to the close proximity of the mother and baby [11, 35, 36].

It is not surprising that the practice of KMC was higher among children with low birth weight compared to those with extremely low birth weight. ELBW infants are typically medically fragile and more vulnerable to complications–they may therefore require specialized care, which may include respiratory, nutritional, and neurodevelopmental support and temperature regulation. These may limit the immediate initiation and practice of KMC in the early stages after birth [16, 24].

## Community- and country-level factors

Surprisingly, mothers from communities with high illiteracy reported higher practice of KMC than those from lower illiteracy communities. Although this finding may seem strange, it is plausible that women resident in areas with greater degrees of illiteracy may have limited exposures to alternative methods of infant care for preterm babies. Since they lack access to formal schooling or written information, they could rely on folklore and local knowledge, which might involve promoting KMC [4, 22, 37]. Knowledge about KMC may also spread though informal channels and community networks, especially from their mothers, grandmothers, and experienced caregivers. It is also plausible that some educated women reside in communities with lower illiteracy in a bid to reduce the cost associated with living in urban areas [22, 23].

As expected, KMC was lower among women residing in communities with high poverty. Poverty often correlates with limited education, information, and access to healthcare services, including prenatal and neonatal care which may impact the ability of women to practice KMC [3, 4, 17]. Women facing economic challenges may have difficulty accessing hospitals or medical facilities where KMC is typically promoted and practiced. The cost of implementing KMC, even if it is a relatively low-cost intervention, can still be a barrier for women with high poverty [9]. By extension, on country-level factors, KMC practice was higher among women resident in countries classified as higher income countries. Higher income countries generally have more developed healthcare systems with better access to prenatal and neonatal care [37, 38].

This disparity in country-level practice of KMC may be attributed to the lack of necessary training and equipment for healthcare providers to provide skin-to-skin care effectively, limiting the adoption of KMC [4, 9, 38]. Additionally, poverty and lack of health insurance can impede access to healthcare services, making it challenging for parents to provide effective skin-to-skin care for their infants, particularly in regions with high poverty rates, such as Sub-Saharan Africa and Latin America and The Caribbeans.

## Strengths and limitations

The cross-sectional nature of the survey data used in this study precludes establishing a causal relationship. There is a possibility that other contextual factors, which were not captured during the survey, may contribute to the low uptake of KMC. Additionally, this study did not include other factors that are relevant to the practice of KMC, particularly those that relate to the health system and healthcare providers. Nonetheless, the strength of this study lies in the use of nationally representative data from several countries.

## Conclusion

This study has used a large data set to explore the drivers of KMC practice among infants born with LBW and the impact of individual-, community- and country level differences in the practice of KMC across LMIC. Identifying impediments to KMC implantation is critical for supporting women. Low uptake of KMC in countries with limited resources is a concerning issue that requires urgent attention. Increasing awareness, education, and support for mothers

and families to practice KMC, as well as training healthcare practitioners, can lead to better outcomes for newborns and reduce neonatal death.

## Acknowledgments

The authors are grateful to ICF Macro, USA, for granting the authors the request to use the Demographic and Health Survey data.

## Author Contributions

**Conceptualization:** Temitayo Victor Lawal.

**Data curation:** Temitayo Victor Lawal.

**Formal analysis:** Temitayo Victor Lawal, Damilola Israel Lawal, Oluwafemi John Adeleye.

**Methodology:** Temitayo Victor Lawal, Damilola Israel Lawal.

**Resources:** Temitayo Victor Lawal.

**Validation:** Temitayo Victor Lawal, Oluwafemi John Adeleye.

**Visualization:** Temitayo Victor Lawal.

**Writing – original draft:** Temitayo Victor Lawal, Damilola Israel Lawal, Oluwafemi John Adeleye.

**Writing – review & editing:** Temitayo Victor Lawal, Damilola Israel Lawal, Oluwafemi John Adeleye.

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
