## [Decision Letter · Decision Letter 0]

13 Jun 2023

PGPH-D-23-00773

Determinants of Kangaroo Mother Care among Low-Birth-Weight Infants in Low Resource Settings

Dear Dr. Lawal,

Thank you for submitting your manuscript to PLOS Global Public Health. After careful consideration, we feel that it has merit but does not fully meet PLOS Global Public Health’s publication criteria as it currently stands. Therefore, we invite you to submit a revised version of the manuscript that addresses the points raised during the review process.

We look forward to receiving your revised manuscript.

Kind regards,

Jianhong Zhou

Staff Editor

Journal Requirements:

Additional Editor Comments (if provided):

Reviewers' comments:

Reviewer's Responses to Questions

**Comments to the Author**

1. Does this manuscript meet PLOS Global Public Health’s publication criteria? Is the manuscript technically sound, and do the data support the conclusions? The manuscript must describe methodologically and ethically rigorous research with conclusions that are appropriately drawn based on the data presented.

Reviewer #1: Yes

Reviewer #2: Yes

2. Has the statistical analysis been performed appropriately and rigorously?

Reviewer #1: Yes

Reviewer #2: Yes

3. Have the authors made all data underlying the findings in their manuscript fully available (please refer to the Data Availability Statement at the start of the manuscript PDF file)?

Reviewer #1: Yes

Reviewer #2: Yes

4. Is the manuscript presented in an intelligible fashion and written in standard English?

Reviewer #1: Yes

Reviewer #2: Yes

5. Review Comments to the Author

Reviewer #1: This is a nice secondary analysis of an existing survey addressing important issues towards KMC.

The literature review:

Please include KMC decreases sepsis.

The methodology and results are well described.

The different models are confusing and not sure if it adds additional value to the results section. The final model is probably the most important.

The discussion requires some reworking.

Please be consistent LBW < 2500g

Line 274; doesn't fit in the paragraph. Please remove

The discussion is all over the place and has lots of repetition.

I suggest dividing the discussion into:

Country level

Individual level

Community level

Then discuss a few important findings under each section.

Don’t repeat in each paragraph that KMC uptake is low, rather address why it is low and how can the uptake be improved. Some of the issues are appropriately mentioned.

At a country level, why is KMC uptake low? Are all the determinants just pooled, put on a hierarchy level, and not individualized to each country?

Several statements in the discussion require references.

Line 359; should be a new subheading on the limitations and strengths of the study.

Reviewer #2: 1. Does this manuscript meet PLOS Global Public Health’s publication criteria?

Yes it does compile to the publication criteria.

2. Has the statistical analysis been performed appropriately and rigorously?

Yes the statistical analysis is done rigorously. The methodology is clear and replicable.

3. Have the authors made all data underlying the findings in their manuscript fully available?

Yes, they have provided links to the data sources.

4. Is the manuscript presented in an intelligible fashion and written in standard English?

Yes, it is clearly written with good flow.

Additional comments:

- The conclusion section of the Abstract Should be rephrased in a way to reflect the prominent finding of the study. The study is about finding the determinant factors for KMC practice.

- On the study design and data section- the sampling technique should be elaborated more.

- Result section, line 187. 68.8% experienced KMC. Is it the standard KMC or does it also include the first hour STS contact that is provided to all newborns?

- Result, Table 1, column 4 & 5. KMC done but no touching of bare skin. there should be STS contact to say KMC done. This should be rephrased as it can be miss leading.

6. PLOS authors have the option to publish the peer review history of their article (what does this mean?). If published, this will include your full peer review and any attached files.

**Do you want your identity to be public for this peer review?** For information about this choice, including consent withdrawal, please see our Privacy Policy.

Reviewer #1: **Yes: **Dr Tanusha Ramdin

Reviewer #2: No

---

## [Editor Report · Decision Letter 1]

11 Aug 2023

PGPH-D-23-00773R1

Determinants of Kangaroo Mother Care among Low-Birth-Weight Infants in Low Resource Settings

Dear Author

Thank you for submitting your manuscript to PLOS Global Public Health. After careful consideration, we feel that it has merit but does not fully meet PLOS Global Public Health’s publication criteria as it currently stands. Therefore, we invite you to submit a revised version of the manuscript that addresses the points raised during the review process.

Please can you include authors response to thereviewe's question to the methods or limitations section:

Result section,  Is it the standard KMC or does it also include the first hour STS contact that is provided to all newborns?

Result, Table 1, column 4 & 5. KMC done but no touching of bare skin. there should be STS contact to say KMC done. This should be rephrased as it can be miss leading.

Response: The authors would like to point out that in this context, it Skin-to-skin (STS) may be slightly different from Kangaroo Mother Care. This is because it is possible to practice KMC with the mother still wearing clothes. Therefore, the variable captured the possibility of KMC practice but an “incomplete” practice. However, in the multivariable analysis, we defined those whose infants did not touch their skin as though they did not practice KMC

We look forward to receiving your revised manuscript.

Kind regards,

Tanusha Ramdin, MMBCH, DCH(SA), FCPAEDS(SA), Mmed(WITS), Neonate

Guest Editor
---

## [Editor Report · Decision Letter 2]

16 Aug 2023

Determinants of Kangaroo Mother Care among Low-Birth-Weight Infants in Low Resource Settings

PGPH-D-23-00773R2

Dear Author

We are pleased to inform you that your manuscript 'Determinants of Kangaroo Mother Care among Low-Birth-Weight Infants in Low Resource Settings' has been provisionally accepted for publication in PLOS Global Public Health.

Best regards,

Tanusha Ramdin, MMBCH, DCH(SA), FCPAEDS(SA), Mmed(WITS), Neonate

Guest Editor